# Dose–Response Analysis of the Tubular and Glomerular Effects of Chronic Exposure to Environmental Cadmium

**DOI:** 10.3390/ijerph191710572

**Published:** 2022-08-25

**Authors:** Soisungwan Satarug, David A. Vesey, Glenda C. Gobe

**Affiliations:** 1Kidney Disease Research Collaborative, Translational Research Institute, Brisbane 4102, Australia; 2Department of Nephrology, Princess Alexandra Hospital, Brisbane 4075, Australia; 3School of Biomedical Sciences, The University of Queensland, Brisbane 4072, Australia; 4NHMRC Centre of Research Excellence for CKD QLD, UQ Health Sciences, Royal Brisbane and Women’s Hospital, Brisbane 4029, Australia

**Keywords:** β_2-_microglobulin, cadmium, creatinine clearance, creatinine excretion, glomerular filtration rate, N-acetyl-β-D-glucosaminidase, tubular dysfunction, toxicity threshold level

## Abstract

We retrospectively analyzed data on the excretion of cadmium (E_Cd_), β_2_-microglobulin (E_β2M_) and N-acetyl-β-D-glucosaminidase (E_NAG_), which were recorded for 734 participants in a study conducted in low- and high-exposure areas of Thailand. Increased E_β2M_ and E_NAG_ were used to assess tubular integrity, while a reduction in the estimated glomerular filtration rate (eGFR) was a criterion for glomerular dysfunction. E_Cd_, E_β2M_ and E_NAG_ were normalized to creatinine clearance (C_cr_) as E_Cd_/C_cr_, E_β2M_/C_cr_ and E_NAG_/C_cr_ to correct for interindividual variation in the number of surviving nephrons and to eliminate the variation in the excretion of creatinine (E_cr_). For a comparison, these parameters were also normalized to E_cr_ as E_Cd_/E_cr_, E_β2M_/E_cr_ and E_NAG_/E_cr_. According to the covariance analysis, a Cd-dose-dependent reduction in eGFR was statistically significant only when E_cd_ was normalized to C_cr_ as E_Cd_/C_cr_ (*F* = 11.2, *p* < 0.001). There was a 23-fold increase in the risk of eGFR ≤ 60 mL/min/1.73 m^2^ in those with the highest E_Cd_/C_cr_ range (*p* = 0.002). In addition, doubling of E_Cd_/C_cr_ was associated with lower eGFR (β = −0.300, *p* < 0.001), and higher E_NAG_/C_cr_ (β = 0.455, *p* < 0.001) and E_β2M_/C_cr_ (β = 0.540, *p* < 0.001). In contrast, a covariance analysis showed a non-statistically significant relationship between E_Cd_/E_cr_ and eGFR (*F* = 1.08, *p* = 0.165), while the risk of low eGFR was increased by 6.9-fold only among those with the highest E_Cd_/E_cr_ range. Doubling of E_Cd_/E_cr_ was associated with lower eGFR and higher E_NAG_/E_cr_ and E_β2M_/E_cr_, with the β coefficients being smaller than in the C_cr_-normalized dataset. Thus, normalization of Cd excretion to C_cr_ unravels the adverse effect of Cd on GFR and provides a more accurate evaluation of the severity of the tubulo-glomerular effect of Cd.

## 1. Introduction

Cadmium (Cd) is a toxic metal with continuing public health concern worldwide due to its toxicity to the mitochondrion, which is central to the maintenance of cell integrity and function [1,2,3]. Environmental exposure to Cd is inevitable for most people because the metal is present in almost all food types [4]. The prominence of Cd in wheat and rice-based products, as recorded by total diet studies undertaken at multiple sites and countries, is particularly concerning, as these food staples are consumed in some form by the majority of the world’s population [4]. The realization in the 1940s that the condition referred to as “itai-itai” disease was due to consumption of rice heavily contaminated with Cd brought into focus the real threat to health posed by this metal [5,6]. Itai-itai disease is the most severe form of human Cd poisoning, characterized by severe damage to the kidneys and bones, resulting in multiple bone fractures due to osteoporosis and osteomalacia [5,6].

A protein β_2_-microglobulin (β_2_M), with a molecular weight of 11,800 Da, is synthesized and shed by all nucleated cells in the body [7]. By virtue of its small mass, β_2_M is filtered freely by the glomeruli and is reabsorbed almost completely by the kidney’s tubular cells [8]. Cd has been shown to cause a reduction in the tubular maximum reabsorption of β_2_M [9], and increased β_2_M excretion has been used as an indicator of impaired tubular reabsorptive function for many decades. An increase in the excretion of β_2_M above 300 μg/g creatinine is used as an endpoint in health risk assessments of Cd in the human diet, and urinary Cd excretion levels below 5.24 µg Cd/g creatinine have been identified as body burdens that are not associated with such an increase in β_2_M excretion [10,11,12]. However, our previous assessment showed that β_2_M excretion of 100–299, 300–999 and ≥ 1000 μg/g creatinine were associated with 4.7-, 6.2- and 10.5-fold increases in the risk of an estimated glomerular filtration rate (eGFR) ≤ 60 mL/min/1.73 m^2^, which is commensurate with chronic kidney disease (CKD) [13]. Thus, elevated β_2_M excretion does not appear to be an early sign of nephrotoxicity to Cd, and the utility of β_2_M excretion as a toxicity criterion of Cd is questionable.

Current epidemiologic evidence suggests that environmental exposure to Cd may reduce the GFR at body burdens much lower than those associated with Cd excretion of 5.24 µg/g creatinine [14,15,16,17]. Reductions in GFR due to Cd nephropathy have often been attributed to glomerular injury. However, sufficient tubular injury can indeed disable glomerular filtration and ultimately leads to nephron atrophy, glomerulosclerosis, and interstitial inflammation and fibrosis [18]. Current evidence also suggests that a common practice of normalizing Cd excretion (E_Cd_) to creatinine excretion (E_cr_) underestimate the severity of Cd toxicity in kidneys, while normalizing E_Cd_ to creatinine clearance (C_cr_) could provide a more accurate measure of the nephrotoxicity of Cd [19,20].

The present study had two major aims: firstly, to ascertain the associations of Cd exposure with the risks of adverse effects on kidney glomerular and tubular function, and, secondly, to compare the impacts of normalizing E_Cd_, E_β2M_ and E_NAG_ to E_cr_ and C_cr_ on estimations of Cd toxicity in the kidneys. Kidney dysfunction was indicated by a reduction in eGFR and increased E_β2M_ and E_NAG_.

## 2. Materials and Methods

### 2.1. Study Subjects

We assembled archived data from 289 men and 445 women who were drawn from Bangkok (a low-exposure area) between 2001 and 2003 and from subsistence farming areas of Mae Sot District (a high-exposure area) in Thailand between 2004 and 2005 [21]. A wide range of Cd exposure levels among the participants enabled an evaluation of the dose–response relationships. The study protocol was approved by the Institutional Ethical Committee of Chulalongkorn University and the Mae Sot Hospital Ethical Committee. At the time of recruitment, all participants had lived at their current addresses for at least 30 years, and all gave informed consent to participate. Exclusion criteria were pregnancy, breastfeeding, a history of metalwork, and a hospital record or physician’s diagnosis of an advanced chronic disease. Smoking, diabetes, hypertension, regular use of medications, educational level, occupation and family health history were ascertained by a questionnaire. Diabetes was defined as fasting plasma glucose levels ≥ 126 mg/dL or a physician’s prescription of anti-diabetic medications. Hypertension was defined as systolic blood pressure ≥ 140 mmHg, diastolic blood pressure ≥ 90 mmHg, and/or a physician’s diagnosis and prescription of anti-hypertensive medications.

### 2.2. Collection of Biological Specimens and Analyses

Second morning urine samples were collected after an overnight fast, and whole blood samples were obtained within 3 hours after the urine samples were collected. The simultaneous collection of blood and urine samples was required to normalize the excretion of Cd, β_2_M and NAG to C_cr_. Aliquots of urine, whole blood and plasma were stored at −20 °C or −80 °C for later analysis. The assay for urine and plasma concentrations of creatinine ([cr]_u_ and [cr]_p_) was based on the Jaffe reaction. The urinary NAG assay was based on colorimetry (NAG test kit, Shionogi Pharmaceuticals, Sapporo, Japan). The urinary β_2_M assay was based on the latex immunoagglutination method (LX test, Eiken 2MGII; Eiken and Shionogi Co., Tokyo, Japan).

For the group from the area polluted by Cd, [Cd]_u_ was determined by atomic absorption spectrophotometry (Shimadzu Model AA-6300, Kyoto, Japan). Urine standard reference material No. 2670 (National Institute of Standards, Washington, DC, USA) was used for quality assurance and control purposes. None of the urine samples from this group was found to have a [Cd]_u_ below the detection limit.

For the group from the low-exposure area, [Cd]_u_ was determined by inductively-coupled plasma mass spectrometry (ICP/MS, Agilent 7500, Agilent Technologies, Santa Clara, CA, USA) because it had the high sensitivity required to measure very low Cd concentrations. Multi-element standards (EM Science, EM Industries, Inc., Newark, NJ, USA) were used to calibrate the Cd analyses. The accuracy and precision of those analyses were ascertained with reference urine (Lyphochek^®^, Bio-Rad, Sydney, Australia). The Cd concentration assigned to samples with Cd below the detection limit was the detection limit of 0.05 µg/L divided by the square root of 2 [22].

### 2.3. Estimated Glomerular Filtration Rate (eGFR)

In theory, the GFR reflects the number of surviving nephrons × the average GFR per nephron, and is indicative of nephron function [23,24]. In practice, the GFR is estimated from various equations, and is reported as eGFR. We used CKD-EPI equations to calculate the eGFR values because they are considered to be the most accurate equations and have been validated with inulin clearance [25]. 

Male eGFR = 141 × [serum creatinine/0.9]^y^ × 0.993^age^, where y = −0.411 if serum creatinine ≤ 0.9 mg/dL or −1.209 if serum creatinine > 0.9 mg/dL.

Female eGFR = 144 × [serum creatinine/0.7]^y^ × 0.993^age^, where y = −0.329 if serum creatinine ≤ 0.7 mg/dL or −1.209 if serum creatinine > 0.7 mg/dL.

CKD was defined as eGFR ≤ 60 mL/min/1.73 m^2^, and CKD Stages 1, 2, 3a, 3b, 4 and 5 corresponded to eGFR values of 90–119, 60–89, 45–59, 30–44, 15–29 and <15 mL/min/1.73 m^2^, respectively [23].

### 2.4. Normalization of Excretion of Cd, β2M and NAG to Ecr and Ccr

E_x_ was normalized to E_cr_ as [x]_u_/[cr]_u_, where x = Cd, β_2_M or NAG; [x]_u_ = urine concentration of x (mass/volume); and [cr]_u_ = urine creatinine concentration (mg/dL). The ratio [x]_u_/[cr]_u_ was expressed in μg/g of creatinine.

E_x_ was normalized to C_cr_ as E_x_/C_cr_ = [x]_u_[cr]_p_/[cr]_u_, where x = Cd, β_2_M or NAG; [x]_u_ = urine concentration of x (mass/volume); [cr]_p_ = plasma creatinine concentration (mg/dL); and [cr]_u_ = urine creatinine concentration (mg/dL). E_x_/C_cr_ was expressed as the excretion of x per volume of filtrate [26]. It is noteworthy that simultaneous collection of blood and urine samples is a prerequisite for C_cr_ normalization.

### 2.5. Statistical Analysis

Data were analyzed with IBM SPSS Statistics 21 (IBM Inc., New York, NY, USA). The distributions of eGFR and the excretion of Cd, β_2_M and NAG were examined for skewness, and those showing rightward skewing were subjected to logarithmic transformation before analysis. The departure of a given variable from a normal distribution was assessed with the one-sample Kolmogorov–Smirnov test. The Mann–Whitney U-test was used to compare mean differences between two groups. The Chi-square test was used to determine differences in percentage and prevalence data. A multivariable regression model analysis was used to evaluate the associations of Cd excretion rates and the dependent variables, which included eGFR and excretion of β_2_M and NAG. Multivariable logistic regression analysis was used to determine the prevalence odds ratio (POR) for dichotomized outcomes. The mean eGFR values adjusted for covariates and their interaction in groups of subjects were obtained by univariate/covariance analysis with Bonferroni correction in multiple comparisons. For each analysis, *p*-values ≤ 0.05 for two-tailed tests were assumed to indicate statistical significance.

## 3. Results

### 3.1. Demographic Characteristics of the Study Subjects

As the data in Table 1 indicate, 27.2% of the subjects included in the present analysis were residents of a low-exposure area.

The mean age of men and women was similar, and the overall mean age was 48.1 years. Smoking status was classified as those who were current smokers and those who had stopped smoking for less than 10 years. The percentage of smokers was higher in men (69.5%) than in women (22.5%), but the percentage of those with diabetes and low eGFR in men and women did not differ. Mean E_Cd_ in smokers was 2.4-fold higher than that of non-smokers (3.80 vs. 1.63 µg/L, *p* < 0.001). The mean plasma and urinary creatinine concentrations were higher in men than in women (*p* < 0.001). Conversely, the mean urinary NAG concentrations were higher in women than in men (*p* = 0.045). The mean urinary concentrations of β_2_M and Cd in men and women did not differ. The mean E_β2M_/E_cr_, mean E_NAG_/E_cr_ and mean E_Cd_/E_cr_ were higher in women than in men (*p* < 0.05). In contrast, the mean E_β2M_/C_cr_, mean E_NAG_/C_cr_ and mean E_Cd_/C_cr_ in women and men did not differ.

### 3.2. Effects of Residential Location and Smoking Habit on Cadmium Excretion Levels

As the data in Table 2 indicate, mean age and the percentages of smokers and those with hypertension among residents of a Cd-contaminated area were all higher than in those who lived in Bangkok. None of the residents of a low-exposure area had diabetes or low eGFR. Half and 23.5% of the subjects from the high- and low-exposure areas were smokers, respectively. The mean E_Cd_ for non-smokers in the high-exposure area was 24.9-fold higher than that of non-smokers from a low-exposure area (5.22 vs. 0.21 µg/L, *p* <0.001). The mean E_Cd_ for smokers from a high-exposure area was higher than that of smokers from a low-exposure area by 18.3-fold (5.87 vs. 0.32 µg/L, *p* <0.001).

The mean BMI in the low- and high-exposure groups was similar (*p* = 0.703). In contrast, the mean eGFR was 18.6% higher in the low-exposure group compared with the high-exposure group (*p* < 0.001). The means of all other measured continuous variables were higher in residents of a Cd-contaminated area than in those who lived in a low-exposure area.

### 3.3. Associations of E_Cd_/E_cr_ vs. E_Cd_/C_cr_ with eGFR, β_2_M and NAG

As the data in Table 3 indicate, a set of seven independent variables, including age, E_Cd_/E_cr_, BMI, sex, hypertension, Type 2 diabetes and smoking, were examined for their associations with the three markers of kidney effects: eGFR deterioration, and increases in β_2_M and NAG excretion.

In the regression model analysis (Table 3), age, E_Cd_/E_cr_, sex, hypertension, diabetes and smoking accounted for 48.5% of the variation in eGFR (*p* < 0.001), 22.2% of the variation in E_NAG_/E_cr_ (*p* < 0.001) and 34.7% of the variation in E_β2M_/E_cr_ (*p* < 0.001). Doubling of E_Cd_/E_cr_ was associated with lower eGFR values (β = −0.142, *p* < 0.001), higher E_NAG_/E_cr_ (β = 0.390, *p* < 0.001) and higher E_β2MG_/E_cr_ (β = 0.499, *p* < 0.001).

In an equivalent regression model analysis with the excretion data normalized to C_cr_ (Table 4), age, E_Cd_/C_cr_, sex, hypertension, diabetes and smoking accounted for 52.9% of the variation in eGFR (*p* < 0.001), 25.7% of the variation in E_NAG_/C_cr_ (*p* < 0.001) and 38.3% of the variation in E_β2M_/C_cr_ (*p* <0.001). Doubling of E_Cd_/C_cr_ was associated with lower eGFR values (β = −0.300, *p* < 0.001), higher E_NAG_/C_cr_ (β = 0.455, *p* < 0.001) and higher E_β2M_/C_cr_ (β = 0.540, *p* < 0.001).

### 3.4. Associations of Cadmium Excretion with Elevated Risks of Nephrotoxicity

Through a logistic regression analysis (Table 5), Cd-dose-dependent effects were seen between E_Cd_/E_cr_ and the risks of low eGFR, E_NAG_/E_cr_ ≥ 4 units/g creatinine and E_β2M_/E_cr_ ≥ 300 µg/g creatinine. A risk of low eGFR was increased by 6.91-fold when the E_Cd_/E_cr_ level rose to 5.77 µg/g creatinine or higher (*p* = 0.011).

E_Cd_/E_cr_ 1.84–5.76 µg/g creatinine was associated with 2.38- and 5.41-fold increases in the risk of abnormal E_NAG_/E_cr_ (*p* < 0.001) and E_β2M_/E_cr_ (*p* <0.001), respectively.

In an equivalent logistic regression analysis with the excretion data normalized to C_cr_ (Table 6), Cd-dose-dependent effects were seen between Cd and the risks of low eGFR, E_NAG_/C_cr_ ≥ 4 units/L filtrate and E_β2M_/C_cr_ ≥ 300 µg/L filtrate. A risk of low eGFR was increased by 23-fold at E_Cd_/C_cr_ levels ≥ 44.6 ng/L filtrate (*p* = 0.002). E_Cd_/C_cr_ 19.5–44.5 ng/L filtrate was associated with 2.69- and 2.84-fold increases in the risk of abnormal E_NAG_/C_cr_ (*p* < 0.001) and E_β2M_/C_cr_ (*p* <0.001), respectively.

### 3.5. Comparing eGFR Reductions and Increases in E_NAG_ and E_β2M_ among Subjects

To further evaluate the effect of normalizing the excretion of Cd, NAG and β_2_M to E_cr_ or C_cr_, we compared the mean eGFR, mean NAG excretion and mean β_2_M excretion together with the variances (the upper and lower bounds of the 95% confidence interval for each mean) in groups of subjects stratified by E_Cd_/E_cr_ or E_Cd_/C_cr_ levels. We used a univariate analysis of variance to derive those figures and, with full factorial models, the means were adjusted for covariates that included age, BMI, diabetes, sex, smoking, hypertension and their interactions. Data from 709 subjects were analyzed because the BMI data for 25 subjects were missing.

E_Cd_/E_cr_ Levels 1, 2 and 3 correspond to E_Cd_/E_cr_ 0.04–1.83, 1.84–5.76 and 5.77–57.7 µg/g creatinine, respectively. The number of subjects with E_Cd_/E_cr_ Levels 1, 2 and 3 was 221, 239 and 249, respectively. E_Cd_/C_cr_ Levels 1, 2 and 3 correspond to E_Cd_/C_cr_ 0.3–19.4, 19.6–44.5 and 44.6–800 ng/L filtrate, respectively. The number of subjects with E_Cd_/E_cr_ Levels 1, 2 and 3 was 265, 174 and 270, respectively.

#### 3.5.1. Cadmium-Dose-Dependent Reductions in eGFR

As Figure 1a,b indicates, the mean eGFRs for each E_Cd_/E_cr_ level were not statistically different (*p* = 0.165). In contrast, the mean eGFR was the highest for E_Cd_/C_cr_ Level 1, intermediate for Level 2 and lowest for Level 3 (*p* < 0.001) (Figure 1c). This Cd-dose-related eGFR reduction was evident in men and women (Figure 1d).

#### 3.5.2. Cadmium-Dose-Dependent Increases in NAG Excretion

The mean NAG excretion was highest at E_Cd_/E_cr_ Level 1, intermediate at Level 2 and lowest at Level 3 (*p* = 0.001) (Figure 2a). In men and women, the mean E_β2M_/E_cr_ was higher in those with E_Cd_/E_cr_ Level 3, compared with E_Cd_/E_cr_ Level 1 (Figure 2b). Similarly, the mean E_NAG_/E_cr_ was lowest at E_Cd_/C_cr_ Level 1, intermediate at Level 2 and highest at Level 3 (*p* < 0.001) (Figure 2c), and the mean E_NAG_/E_cr_ was higher in men and women who had E_Cd_/C_cr_ Level 3, compared those who had E_Cd_/C_cr_ Level 1 (Figure 2d).

#### 3.5.3. Cadmium-Dose-Dependent Increases in β_2_M Excretion

The mean E_β2M_/E_cr_ was highest at E_Cd_/E_cr_ Level 1, intermediate at Level 2 and lowest at Level 3 (*p* < 0.001) (Figure 3a). In men, the mean E_β2M_/E_cr_ was higher in those with E_Cd_/E_cr_ Levels 2 and 3, compared with E_Cd_/E_cr_ Level 1 (Figure 3b). In women, the mean E_β2M_/E_cr_ at Level 3 of E_Cd_/E_cr_ was higher than at Level 1 (Figure 3b). Similarly, the mean E_β2M_/E_cr_ was highest at E_Cd_/C_cr_ Level 1, intermediate at Level 2 and lowest at Level 3 (*p* < 0.001) (Figure 3c), and the mean E_β2M_/E_cr_ was higher in men with E_Cd_/C_cr_ Levels 2 and 3, compared with Level 1 (Figure 3d). In women, only the mean E_β2M_/E_cr_ at Level 3 was higher than at Level 1 (Figure 3d).

## 4. Discussion

Normalization of urinary concentrations of excreted Cd, excreted β_2_M and excreted NAG to C_cr_ strengthened the associations of Cd with all three indicators of kidney dysfunction, decreased eGFR, and increased E_β2M_ and E_NAG_. Normalization by C_cr_ increased the proportions of accountable variations in all three effect indicators with adjustment for a set of covariates, including age and BMI (Table 3 vs. Table 4). Because normalizing E_Cd_ and other nephrotoxic indicators to C_cr_ corrected for interindividual variation in the number of surviving nephrons and eliminated the variation in E_cr_, the severity of Cd nephrotoxicity or the effect size of Cd could be assessed with a sufficiently high degree of statistical certainty.

In comparison, although normalization by E_cr_ corrected for urine dilution, this practice introduced an additional variation that was unrelated to Cd exposure or GFR. For example, E_cr_ in women was universally lower than in men due to their lower muscle mass, which is a determinant of E_cr_, and E_Cd_/E_cr_ was consequently higher in women than in men. As the data in Table 1 indicate, the mean E_Cd_/E_cr_, the mean E_β2MG_/E_cr_ and the mean E_NAG_/E_cr_ were all higher in women than in men. In contrast, neither of these parameters was statistically different when excretion was normalized to C_cr_.

A notable effect of adding the variance in the dataset with the conventional method of normalizing E_Cd_, E_β2M_ and E_NAG_ to E_cr_ is demonstrable by covariance analysis. The effect on eGFR was insignificant when [Cd]_u_ was normalized to E_cr_ (Figure 1a,b). In contrast, a Cd-dose-dependent reduction in eGFR was seen in all subjects, men and women when E_Cd_ was normalized to C_cr_ (Figure 1c,d). In addition, there was a 23-fold increase in the risk of low eGFR among subjects with E_Cd_/C_cr_ at the highest level (≥44.6 ng/L filtrate, equivalent to ≥0.0446 µg/L filtrate) in a logistic regression analysis (Table 6). Only a 6.9-fold increase in the risk of low eGFR was seen among those with E_Cd_/E_cr_ at the highest level (≥5.77 µg/g creatinine) (Table 5).

As a consequence of the normalization of E_Cd_ to E_cr_, the impact of Cd on GFR was not realized. In a systematic review and meta-analysis of data from 28 studies [23], a 1.35-fold increase in the risk of proteinuria was seen when comparing the highest vs. lowest category of Cd dose metrics, while an increase in the risk of a reduced eGFR was statistically insignificant (*p* = 0.10). The statistically non-significant association between E_Cd_ and eGFR was likely due to a confounding effect of normalizing urinary Cd levels to the excretion of creatinine as E_Cd_/E_cr_, which was used by studies included in the meta-analysis by Jalili et al. [27]. An erroneous conclusion that Cd was not associated with a progressive reduction in GFR was also made in another systematic review by Byber et al. [28].

Our previous quantitative analysis of E_Cd_ in relation to E_NAG_ suggested that Cd inflicts tubular cell injury at low intracellular concentrations, and that the toxicity intensifies as the concentration of Cd rises [19]. Inflammation and fibrosis follow, nephrons are lost, and the GFR falls [19]. In a recent histopathological examination of kidney biopsies from healthy kidney transplant donors [29], the degree of tubular atrophy was positively associated with the level of Cd accumulation. Tubular atrophy was observed at relatively low Cd levels (median: 13 µg/g wet tissue weight) [29].

The results of the present study confirm an inverse association between E_Cd_ and eGFR, which became apparent only after normalizing E_Cd_ to C_cr_, as reported previously by us [20]. They also confirm a positive association between E_Cd_ and E_NAG_, an indicator of the injury to tubular cells by Cd. However, the effect of Cd accumulation in tubular cells on tubular reabsorptive dysfunction, indicated by an increase in E_B2M_, was not examined in our previous study. In effect, the present work provides evidence for the concurrent effects of chronic Cd exposure on eGFR, tubular injury and impaired tubular reabsorptive function in a Cd-dose-dependent manner.

It is noteworthy that E_Cd_ has been associated with tubular dysfunction in numerous studies, but only a few studies have simultaneously considered the tubular and glomerular effects of Cd. For example, in a study of 208 Guatemalan sugarcane cutters, higher E_Cd_ was associated with lower eGFR (β: −4.23) and higher excretion of neutrophil gelatinase-associated lipocalin, another marker of kidney tubular cell injury (β: 2.92) [30]. Both tubular and glomerular effects were observed in a study of Swedish women, where urinary Cd 0.67 µg/g creatinine was positively associated with E_NAG_ while E_Cd_ 0.87 µg/g creatinine was inversely associated with eGFR [31].

Like NAG, kidney injury molecule 1 (KIM1), which is detectable in urine, originates from the tubular cells, and its excretion is correlated with that of Cd [32]. Excretion of KIM1 (E_KIM1_) has been used to quantify the injury to kidney tubular cells associated with Cd exposure [32,33,34]. KIM1 is found in the urine only after tubular injury has occurred [32,35]. An association between the excretion of Cd and KIM1 was noted in Taiwanese subjects with CKD after adjusting for covariates [16]. No correlation was found between excretion of Cd and protein, thereby suggesting that urinary KIM1 levels could serve as an early warning sign of kidney injury due to low-dose Cd exposure. In a cross-sectional study of 260 men and 440 women who were residents of a Cd-contaminated area of Thailand, E_KIM1_ was found to be more sensitive than E_NAG_ and E_β2MG_ for assessing the tubular effects of Cd [36].

Low environmental Cd exposure has consistently been associated with a reduction in eGFR among participants in various National Health and Nutrition Examination Survey cycles undertaken in the U.S. between 1999 and 2016 [14,37,38,39]. The geometric mean, and the 50th, 75th, 90th and 95th percentile values for urinary Cd levels in the representative U.S. general population were 0.210, 0.208, 0.412, 0.678 and 0.949 µg/g creatinine, and the corresponding values for blood Cd were 0.304, 0.300, 0.500, 1.10 and 1.60 µg/L, respectively [40]. According to these figures, environmental Cd exposure levels in the U.S. could be considered to be low. Cd exposure, measured as E_Cd_, was associated with eGFR reductions in studies from Taiwan, Myanmar and Thailand [16,17,41].

Blood Cd levels of 1.74 μg/L and 2.08 μg/L were associated with low eGFR in two Korean population studies [42,43]. These blood Cd levels were approximately fourfold higher than the blood Cd levels found to be associated with low eGFR in a representative U.S. population [37,38,39]. An inverse association was seen between blood Cd and eGFR in another study of 1984 Koreans aged ≥19 years [44]. Although studies from various countries report disparate levels of environmental exposure to Cd, they are broadly consistent in that they have found that the exposure levels associated with an increased risk of low eGFR do not exceed a prescribed Cd toxicity threshold level of 5.24 µg/g creatinine. This Cd toxicity threshold level was derived from a risk assessment model that assumes β_2_M excretion above ≥300 µg/g creatinine as an endpoint [[10].[11]].

The data in the present study strongly argue that a Cd-induced reduction in eGFR could be a suitable nephrotoxicity endpoint for health risk calculations. This endpoint is clinically relevant and is more sensitive to Cd toxicity than E_β2M_. In clinical trials, successful treatment of CKD is judged by the attenuation of a decline in eGFR [23,45].

## 5. Conclusions

The impact of Cd exposure on GFR has long been underestimated due to the common practice of normalizing E_Cd_ to E_cr_. Consequently, the effects of environmental exposure to Cd on GFR and CKD risk have not been addressed adequately. The established nephrotoxicity threshold level for Cd is outdated and is not protective of human health. As a starting point, the comparability of guidelines between populations could be improved by universal acceptance of a consistent normalization of excretion of Cd to C_cr_ that circumvents the effect of muscle mass on creatinine excretion and gives a more accurate assessment of Cd nephropathy. It would also be beneficial if a reduction in eGFR was accepted as a critical effect for Cd toxicity suitable for health risk calculations, since reduced eGFR is more clinically relevant and more sensitive than a rise in excreted β_2_M.

## Figures and Tables

**Figure 1 ijerph-19-10572-f001:**
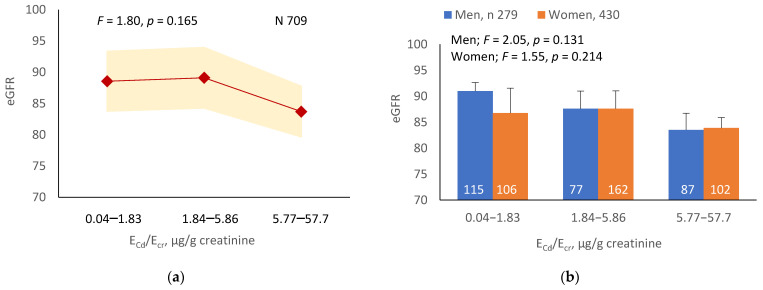
Cadmium-dose-dependent reductions in eGFR. The color-coded area graphs depict eGFR means with variances in subjects with E_Cd_/E_cr_ Levels 1, 2 and 3 (**a**) and in subjects with E_Cd_/C_cr_ Levels 1, 2 and 3 (**c**). The bar graphs show the eGFR means in men and women with E_Cd_/E_cr_ Levels 1, 2 and 3 (**b**) and the eGFR means in men and women with E_Cd_/C_cr_ Levels 1, 2 and 3 (**d**). Where appropriate, statistical comparisons are made within each sex between eGFR means in bars a and b. The GM (SD) values of E_Cd_/E_cr_ at Levels 1, 2 and 3 are 0.51 (0.50), 3.43 (1.10), and 0.94 (7.33) µg/g creatinine, with the corresponding number of subjects being 221, 239, and 249, respectively. The GM (SD) values of E_Cd_/C_cr_ at Levels 1, 2 and 3 are 5.18 (5.72), 29.6 (6.73) and 97.3 (97.7) ng/L filtrate, with the corresponding number of subjects being 265, 174 and 270, respectively.

**Figure 2 ijerph-19-10572-f002:**
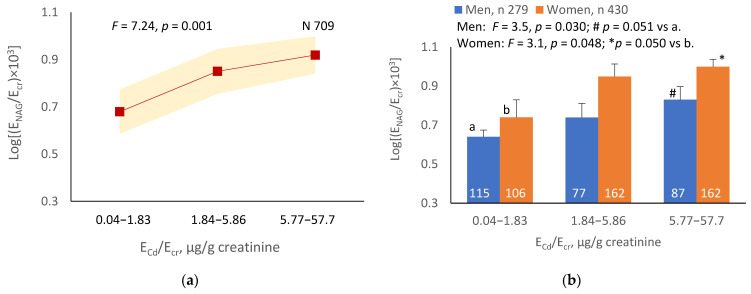
Cadmium-dose-related increases in NAG excretion. The color-coded area graphs depict NAG excretion means as log[(E_NAG_/E_cr_) × 10^3^] values with variances for subjects with E_Cd_/E_cr_ Levels 1, 2 and 3 (**a**) and NAG excretion means as log[(E_NAG_/C_cr_) × 10^4^] with variances in subjects with E_Cd_/C_cr_ Levels 1, 2 and 3 (**c**). The bar graphs show NAG excretion means as log[(E_NAG_/E_cr_) × 10^3^] values in men and women with E_Cd_/E_cr_ Levels 1, 2, and 3 (**b**) and NAG excretion means as log[(E_NAG_/C_cr_) × 10^4^] in men and women with E_Cd_/C_cr_ Levels 1, 2 and 3 (**d**). Where appropriate, statistical comparisons are made within each sex between NAG excretion means in bars a and b. The GM (SD) values of E_Cd_/E_cr_ at Levels 1, 2 and 3 are 0.51 (0.50), 3.43 (1.10) and 10.94 (7.33) µg/g creatinine, with the corresponding number of subjects being 221, 239 and 249, respectively. The GM (SD) values of E_Cd_/C_cr_ at Levels 1, 2 and 3 are 5.18 (5.72), 29.6 (6.73) and 97.3 (97.7) ng/L filtrate, with the corresponding number of subjects being 265, 174 and 270, respectively.

**Figure 3 ijerph-19-10572-f003:**
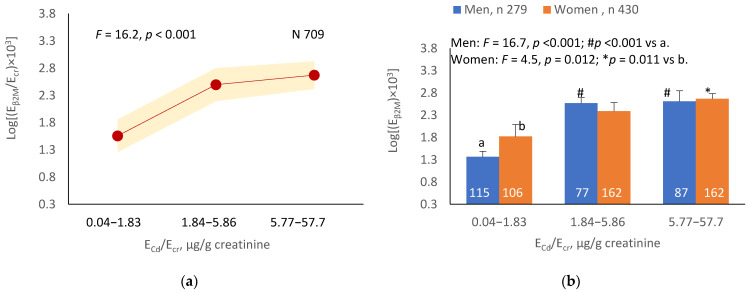
Cadmium-dose-related increases in β_2_MG excretion. The color-coded area graphs depict β_2_M excretion means as log[(E_β2M_/E_cr_) × 10^3^] values with variances for subjects with E_Cd_/E_cr_ Levels 1, 2 and 3 (**a**) and β_2_M excretion means as log[(E_β2M_/C_cr_) × 10^4^] values with variances for subjects with E_Cd_/C_cr_ Levels 1, 2 and 3 (**c**). The bar graphs show β_2_M excretion means as log[(E_β2M_/E_cr_) × 10^3^] values in men and women with E_Cd_/E_cr_ Levels 1, 2 and 3 (**b**) and β_2_M excretion means as log[(E_β2M_/C_cr_) × 10^4^] values in men and women with E_Cd_/C_cr_ Levels 1, 2 and 3 (**d**). Where appropriate, statistical comparisons are made within each sex between mean β_2_M excretion as log[(E_β2M_/E_cr_) × 10^3^] or log[(E_β2M_/C_cr_) × 10^4^] in bars a and b. The GM (SD) values of E_Cd_/E_cr_ at Levels 1, 2 and 3 are 0.51 (0.50), 3.43 (1.10) and 10.94 (7.33) µg/g creatinine, with the corresponding number of subjects being 221, 239 and 249, respectively. The GM (SD) values of E_Cd_/C_cr_ at Levels 1, 2 and 3 are 5.18 (5.72), 29.6 (6.73) and 97.3 (97.7) ng/L filtrate, with the corresponding number of subjects being 265, 174 and 270, respectively.

**Table 1 ijerph-19-10572-t001:** Characteristics of the study subjects, stratified by sex.

Parameters	All Subjects, *n* 734	Males, *n* 289	Females, *n* 445	*p*
From a low-exposure location (%)	27.2	34.6	22.5	<0.001
Smoking (%)	42.8	69.6	25.4	<0.001
Hypertension (%)	31.7	29.1	33.5	0.209
Diabetes (%)	1.5	2.1	1.1	0.299
Age, years	48.1 ± 11.0	47.4 ± 13.5	48.6 ± 9.1	0.059
BMI, kg/m^2^	23.2 ± 3.8	22.4 ± 3.3	23.7 ± 4.0	<0.001
eGFR ^a^, mL/min/1.73 m^2^	91.0 ± 21.7	90.3 ± 22.9	91.4 ± 20.9	0.510
Low eGFR ^b^ (%)	9	10	8.3	0.426
Plasma creatinine, mg/dL	0.85 ± 0.28	1.00 ± 0.30	0.76 ± 0.22	<0.001
Urine creatinine, mg/dL	85.59 ± 73.86	96.93 ± 73.34	78.95 ± 73.42	<0.001
Urine β_2_M, µg/L	112 ± 24,510	98 ± 20,140	122 ± 26,989	0.401
Urine NAG, U/L	5.48 ± 9.96	4.97 ± 8.25	5.84 ± 10.86	0.045
Urine Cd, µg/L	2.34 ± 8.98	2.05 ± 10.87	2.54 ± 7.50	0.248
Normalized to E_cr_ as E_x_/E_cr_ ^c^				
E_β2M_/E_cr_, µg/g creatinine	130 ± 20,620	101 ± 16,333	154 ± 22,997	0.016
E_NAG_/E_cr_, U/g creatinine	6.40 ± 8.95	5.13 ± 4.73	7.39 ± 10.62	<0.001
E_Cd_/E_cr_, µg/g creatinine	2.73 ± 6.61	2.12 ± 6.64	3.22 ± 6.57	<0.001
Normalized to C_cr_ as E_x_/C_cr_ ^d^				
E_β2M_/E_cr_ × 100, µg/L filtrate	110 ± 29,676	101 ± 33,060	118 ± 27,253	0.320
E_NAG_/E_cr_ × 100, U/L filtrate	5.42 ± 9.40	5.11 ± 6.36	5.63 ± 10.91	0.166
E_Cd_/C_cr_ × 100, µg/L filtrate	2.31 ± 7.82	2.11 ± 8.36	2.46 ± 7.44	0.249

*n*, number of subjects; BMI, body mass index; eGFR, estimated glomerular filtration rate; E_x_, excretion of x; cr, creatinine; C_cr_, creatinine clearance; β_2_M, β_2_-microglobulin; NAG, N-acetyl-β-D-glucosaminidase, Cd, cadmium; ^a^ eGFR, was determined by Chronic Kidney Disease Epidemiology Collaboration (CKD−EPI) equations [23]. ^b^ Low eGFR was defined as eGFR ≤ 60 mL/min/1.73 m^2^. ^c^ E_x_/E_cr_ = [x]_u_/[cr]_u_; ^d^ E_x_/C_cr_ = [x]_u_[cr]_p_/[cr]_u_, where x = Cd, β_2_M or NAG [26]. Data for age, eGFR and BMI are arithmetic means ± standard deviation (SD). Data for all other continuous variables are geometric means (GM) ± SD. Data for BMI values are from 709 subjects; data for all other variables are from 734 subjects. *p* ≤ 0.05 identifies statistical significance, determined by Pearson’s Chi-square test for percentage differences and the Mann–Whitney U-test for mean differences.

**Table 2 ijerph-19-10572-t002:** Characteristics of study subjects, stratified by residential location.

Parameters	Low-Exposure Area, *n* 200	High-Exposure Area, *n* 534	*p*
Smoking (%)	23.5	50	<0.001
Diabetes (%)	0	2.1	0.041
Hypertension (%)	19.5	36.3	<0.001
Mean age (range), years	39.3 (16–60)	51.4 (30–87)	<0.001
Mean BMI (range), kg/m^2^	23.2 (13.1–39.0)	23.2 (12.3–38.9)	0.703
Mean eGFR ^a^ (range) mL/min/1.73 m^2^	105.2 (65–138)	85.6 (20–131)	<0.001
Low eGFR ^b^ (%)	0	12.4	<0.001
Plasma creatinine, mg/dL	0.79 ± 0.17	0.87 ± 0.30	<0.001
Urine creatinine, mg/dL	55.0 ± 66.0	101.0 ± 72.6	<0.001
Urine β_2_M, µg/L	4.2 ± 119.3	382 ± 28,605	<0.001
Urine NAG, U/L	3.68 ± 4.01	7.87 ± 9.77	<0.001
Urine Cd, µg/L	0.23 ± 0.58	5.54 ± 9.67	<0.001
Normalized to E_cr_ as E_x_/E_cr_ ^c^			
E_β2M_/E_cr_, µg/g creatinine	7.60 ± 118	379 ± 24,048	<0.001
E_NAG_/E_cr_, U/g creatinine	3.68 ± 4.01	7.87 ± 9.77	<0.001
E_Cd_/E_cr_, µg/g creatinine	0.43 ± 0.44	5.48 ± 6.84	<0.001
Normalized to C_cr_ as E_x_/C_cr_ ^d^			
E_β2M_/E_cr_ × 100, µg/L filtrate	5.97 ± 111	330 ± 34,630	<0.001
E_NAG_/E_cr_ × 100, U/L filtrate	2.90 ± 3.62	6.86 ± 10.42	<0.001
E_Cd_/C_cr_ × 100, µg/L filtrate	0.34 ± 0.38	4.77 ± 8.44	<0.001

*n*, number of subjects; BMI, body mass index; eGFR, estimated glomerular filtration rate; E_x_, excretion of x; cr, creatinine; C_cr_, creatinine clearance; β_2_M, β_2_-microglobulin; NAG, N-acetyl-β-D-glucosaminidase, Cd, cadmium; ^a^ eGFR was determined with CKD−EPI equations [23]. ^b^ Low eGFR was defined as eGFR ≤ 60 mL/min/1.73 m^2^. ^c^ E_x_/E_cr_ = [x]_u_/[cr]_u_; ^d^ E_x_/C_cr_ = [x]_u_[cr]_p_/[cr]_u_, where x = Cd, β_2_M or NAG [26]. Data for age, eGFR and BMI are arithmetic mean (range) values. Data for all other continuous variables are GM ± SD values. Data for BMI values are from 709 subjects; data for all other variables are from 734 subjects. *p* ≤ 0.05 identifies statistical significance, determined by Pearson’s Chi-square test for percentage differences and the Mann–Whitney U-test for mean differences.

**Table 3 ijerph-19-10572-t003:** Associations of the markers of kidney effects with E_Cd_/E_cr_.

Independent Variables/Factors	eGFR ^a^, mL/min/1.73 m^2^		Log[(E_NAG_/E_cr_) × 10^3^], µg/g Creatinine		Log[(E_β2M_/E_cr_) × 10^3^], µg/g Creatinine	
	β Coefficient ^b^	*p*	β Coefficient	*p*	β Coefficient	*p*
Age, years	−0.601	<0.001	−0.085	0.038	0.078	0.038
Log_2_ [(E_Cd_/E_cr_) × 10^3^], µg/g creatinine	−0.142	<0.001	0.390	<0.001	0.499	<0.001
BMI, kg/m^2^	−0.027	0.344	0.083	0.019	−0.040	0.216
Sex	−0.084	0.008	−0.178	<0.001	−0.037	0.291
Hypertension	0.036	0.202	−0.180	<0.001	−0.091	0.004
Diabetes	0.070	0.011	−0.024	0.486	−0.008	0.788
Smoking	0.002	0.954	−0.043	0.273	−0.088	0.015
Adjusted R^2^	0.485	<0.001	0.222	<0.001	0.347	<0.001

Coding: hypertensive = 1; normotensive = 2; diabetic = 1; non-diabetic = 2; male = 1; female = 2; smoker = 1; non-smoker = 2. ^a^ eGFR, estimated glomerular filtration rate; BMI, body mass index. ^b^ Standardized regression coefficients (β) indicate the strength of associations between each dependent variable and an individual independent variable in the first column. *p* ≤ 0.05 identifies statistical significance. Adjusted R^2^ values indicate the total variation in each effect indicator explained by a set of seven independent variables.

**Table 4 ijerph-19-10572-t004:** Associations of the markers of kidney effects with E_Cd_/C_cr_.

Independent Variables/Factors	eGFR ^a^, mL/min/1.73 m^2^		Log[(E_NAG_/C_cr_) × 10^4^], µg/L Filtrate		Log[(E_β2M_/C_cr_) × 10^3^], µg/L Filtrate	
	β Coefficient ^b^	*p*	β Coefficient	*p*	β Coefficient	*p*
Age, years	−0.513	<0.001	−0.012	0.775	0.062	0.100
Log_2_ [(E_Cd_/C_cr_) × 10^5^], µg/L filtrate	−0.300	<0.001	0.455	<0.001	0.540	<0.001
BMI, kg/m^2^	−0.032	0.249	0.086	0.013	−0.038	0.227
Sex	−0.086	0.004	−0.034	0.360	−0.033	0.329
Hypertension	0.032	0.235	−0.182	<0.001	−0.088	0.005
Diabetes	0.055	0.036	−0.040	0.221	−0.006	0.844
Smoking	−0.023	0.455	−0.037	0.338	−0.081	0.021
Adjusted R^2^	0.529	<0.001	0.257	<0.001	0.383	<0.001

Coding: hypertensive = 1; normotensive = 2; diabetic = 1; non-diabetic = 2; male = 1; female = 2; smoker = 1; non-smoker = 2. ^a^ eGFR, estimated glomerular filtration rate; BMI, body mass index. ^b^ Standardized regression coefficients (β) indicate the strength of an association between each effect indicator and an individual independent variable in the first column. *p* ≤ 0.05 identifies statistical significance. Adjusted R^2^ values indicate the total variation in each effect indicator explained by a set of seven independent variables.

**Table 5 ijerph-19-10572-t005:** Dose–response analysis of E_Cd_/E_cr_ and the risk of adverse effects on the kidneys.

Independent Variables/Factors	eGFR ≤ 60 mL/min/1.73 m^2^		E_NAG_/E_cr_ ≥ 4 U/g Creatinine		E_β2M_/E_cr_ ≥ 300 µg/Creatinine	
	POR (95% CI)	*p*	POR (95% CI)	*p*	POR (95% CI)	*p*
Age, years	1.151 (1.112, 1.192)	<0.001	1.022 (1.002, 1.042)	0.033	1.017 (0.999, 1.036)	0.069
BMI, kg/m^2^	1.074 (0.987, 1.168)	0.096	1.012 (0.960, 1.068)	0.654	0.996 (0.950, 1.045)	0.880
Diabetes	2.211 (0.448, 10.92)	0.330	1.121 (0.232, 5.421)	0.887	0.653 (0.190, 2.241)	0.498
Sex	1.140 (0.588, 2.329)	0.719	1.653 (1.070, 2.553)	0.023	1.004 (0.677, 1.489)	0.986
Smoking	0.963 (0.469, 1.976)	0.918	0.926 (0.589, 1.455)	0.739	1.119 (0.760, 1,649)	0.569
Hypertension	1.641 (0.852, 3.160)	0.139	2.808 (1.745, 4.517)	<0.001	1.420 (0.990, 2.034)	0.056
E_Cd_/E_cr_, µg/g creatinine						
0.04–1.83	Referent		Referent		Referent	
1.84–5.76	3.271 (0.705, 15.17)	0.130	2.386 (1.469, 3.873)	<0.001	5.407 (3.281, 8.911)	<0.001
5.77–57.7	6.911 (1.545, 30.91)	0.011	2.400 (1.415, 4.073)	0.001	7.502 (4.440, 12.68)	<0.001

Coding: hypertensive = 1; normotensive = 2; diabetic = 1; non-diabetic = 2; male = 1; female = 2; smoker = 1; non-smoker = 2; POR, prevalence odds ratio; CI, confidence interval. The GM (SD) values of E_Cd_/E_cr_ 0.04–1.83, 1.84–5.76 and 5.77–57.7 µg/g creatinine are 0.51 (0.50), 3.43 (1.10) and 10.94 (7.33) µg/g creatinine with the corresponding number of subjects being 221, 239 and 249, respectively. Data were generated from logistic regression analyses relating the POR for the three indicators of adverse effects to the seven independent variables listed in the first column. *p*-values < 0.05 indicate a statistically significant increase in the POR for adverse effects.

**Table 6 ijerph-19-10572-t006:** Dose–response analysis of E_Cd_/C_cr_ and the risk of adverse effects on the kidneys.

Independent Variables/Factors	eGFR ≤ 60 mL/min/1.73 m^2^		E_NAG_/C_cr_ × 100 ≥ 4 U/L Filtrate		E_β2M_/C_cr_ × 100 ≥ 300 µg/L Filtrate	
	POR (95% CI)	*p*	POR (95% CI)	*p*	POR (95% CI)	*p*
Age, years	1.146 (1.107, 1.188)	<0.001	1.043 (1.023, 1.064)	<0.001	1.028 (1.009, 1.027)	0.003
BMI, kg/m^2^	1.091 (1.000, 1.190)	0.051	1.051 (1.001, 1.104)	0.046	1.011 (0.965, 1.060)	0.641
Diabetes	2.033 (0.417, 9.913)	0.380	3.375 (0.407, 27.99)	0.260	0.684 (0.197, 2.377)	0.550
Sex	1.173 (0.568, 2.425)	0.666	0.765 (0.506, 1.157)	0.204	1.023 (0.691, 1.514)	0.911
Smoking	0.914 (0.440, 1.897)	0.809	1.258 (0.832, 1.904)	0.277	1.246 (0.844, 1.841)	0.269
Hypertension	1.717 (0.883, 3.338)	0.111	2.399 (1.606, 3.582)	<0.001	1.494 (1.042, 2.142)	0.029
E_Cd_/C_cr_, ng/L filtrate						
0.3–19.4	Referent		Referent		Referent	
19.5–44.5	7.814 (0.957, 63.81)	0.055	2.689 (1.671, 4.328)	<0.001	2.842 (1.780, 4.538)	<0.001
44.6–800	23.75 (3.107, 181.6)	0.002	2.016 (1.299, 3.128)	0.002	4.782 (3.045, 7.512)	<0.001

Coding: hypertensive = 1; normotensive = 2; diabetic = 1; non-diabetic = 2; male = 1; female = 2; smoker = 1; non-smoker = 2; POR, prevalence odds ratio; CI, confidence interval. The GM (SD) values of E_Cd_/C_cr_ of 0.3–19.4, 19.6–44.5 and 44.6–800 ng/L filtrate are 5.18 (5.72), 29.6 (6.73) and 97.3 (97.7) ng/L filtrate, with the corresponding number of subjects being 265,174 and 270, respectively. Data were generated from logistic regression analyses relating the POR for the three indicators of adverse effects to the seven independent variables listed in the first column. *p*-values < 0.05 indicate a statistically significant increase in the POR for adverse effects.

## Data Availability

All data are contained within this article.

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
