# Peer review of "Dose–Response Analysis of the Tubular and Glomerular Effects of Chronic Exposure to Environmental Cadmium"

_ijerph, 2022, doi:10.3390/ijerph191710572_

Round 1

Reviewer 1 Report

The introduction referring to itai itai disease could be expanded and give more background data and also mention the studies reported from other countries where toxic effects have been reported for populations.

An overview of cadmium and health effects would help the reader to understand cadmium toxicity of todays knowledge.

The data give percentage of smokers in each group. However, it is important that the authors comment on why they do not report data for smokers, non smokers and stopped smoking at least 10 years before this study was performed since tobacco contributes to the exposure to cadmium.

The terminology is different with regard to critical concentration and other terms. Reference 11 is missing year for publication.

After these comments have been considered the manuscript can be published.

Author Response

Response to Reviewer 1

We thank the Reviewer for his/her evaluation of our paper and for comments. Accordingly, we have revised our paper and our response to comments are provided point-by-point as follows. Changes to the text are in blue.

Point 1

The introduction referring to itai itai disease could be expanded and give more background data and also mention the studies reported from other countries where toxic effects have been reported for populations.

Response: A succinct description of itai-itai disease has been added (lines 40-43).

Point 2

An overview of cadmium and health effects would help the reader to understand cadmium toxicity of todays knowledge.

Response: A new reference reporting an update of Cd toxicity in humans (ref. 3) has been added. We have rewritten part of the Introduction to better reflect the content of our paper.

Ref. 3. Satarug, S.; Gobe, G.C.; Vesey, D.A. Multiple targets of toxicity in environmental exposure to low-dose cadmium. Toxics 2022, 10, 472.

Point 3

The data give percentage of smokers in each group. However, it is important that the authors comment on why they do not report data for smokers, nonsmokers and stopped smoking at least 10 years before this study was performed since tobacco contributes to the exposure to cadmium.

Response: To fully address the issue raised, a new Table 2 has been inserted together with a new subsection 3.2. Effects of Residential Location and Smoking on Cadmium Excretion Levels (lines 167-186).

Point 4

The terminology is different with regard to critical concentration and other terms. Reference 11 is missing year for publication.

Response: We have reworded those terms and use them consistently. We have added two more relevant references to reflect the commonality of the regulatory terms (ref. 7, ref. 8). A year of publication of a referred reference has been added.

After these comments have been considered the manuscript can be published.

Response: We have carefully considered and addressed fully all issues raised by the Reviewer.

Reviewer 2 Report

Review of the manuscript IJERPH-1868859

Dose-Response Analysis of the Tubular and Glomerular Effects of Chronic Exposure to Environmental Cadmium

by Satarug et al.

The evaluated manuscript is about the assessment of cadmium nephrotoxicity based on a retrospective evaluation of patients with chronic low and high exposure to environmental cadmium (Cd). The authors aimed to ascertain a relationship between Cd exposure with risk of adverse effects on kidney glomerular and tubular function and to evaluate the clinical usefulness of the normalization of excretion rates of Cd, beta-2-microglobulin (B2MG) and N-acetyl-beta-D-glucosaminidase (NAG) to excretion of clearance of creatinine in the laboratory assessment of cadmium-induced nephrotoxicity. The authors concluded that normalization of urinary excretion of Cd, B2MG and NAG to creatinine clearance. Normalization of urinary excretion of Cd, β2MG and NAG to Ccr strengthened the associations of Cd with all three effect indictors; eGFR, β2MG and NAG excretion rates. Moreover, the Cd toxicity threshold is based on excretion of B2MB > 300 ug/g creatinine. According to Authors, the better endpoint for the Cd-induced nephrotoxicity and health risk calculation is reduction in GFR.

The description and conclusions relating to the nephrotoxicity of cadmium is nothing new and it has been proven many times in toxicology. The novelty of the manuscript may be the authors' proposal to use certain parameters, especially the urinary cadmium excretion normalized to creatinine clearance in estimation of Cd-reduction of GFR.

Reading the manuscript raised some questions and doubts:

1. Abstract of the paper should mention the methodology of work to a greater extent; one may wrongly get the impression that the results were obtained in a clinical trial and not an analytical, retrospective study.

2. Introduction lines 67-70 - one of the formulas for estimating GFR was mentioned (CKD-EPI formula). At this point, it is also worth mentioning other formulas that enable the estimation of GFR. What about the MDRD formula; why authors decided to use the CKD-EPI formula and not MDRD one?

3. Materials and Methods - reading this section may give rise to the erroneous impression that the authors conducted their own research, while their research is a retrospective study and it is based on a population previously enrolled in other clinical study. Thus, the title of the source study, the years of the project's implementation, and description the study protocol should be provided in more detail at the beginning of this section.

4. Materials and Methods – 2.2. Collection of biological specimens and analyses - urinary cadmium concentrations in the two populations distinguished on the basis of expected cadmium exposure (low vs. high) were determined by two different analytical methods. I wonder why the authors did not measure the cadmium concentration in the urine of patients from the area with high cadmium exposure also using the ICP / MS method, that is regarded to be more much more precise and accurate compared to atomic absorption spectrophotometry?

5. In the Results chapter and in Table 1, the authors presented the summary characteristics of the patients. In my opinion, a comparison of the two subpopulations should also be introduced. The research covered both people living in areas with low and high environmental exposure to cadmium. Thus, both groups should be characterized, not just the aggregate population.

6. Materials and Methods – line 80 - what other drugs (if any) were used by the study subjects, except those ones used to treat hypertension or diabetes?

7. Discussion – lines 318-320 – Authors mentioned neutrophil gelatinase-associated lipocalin NGAL-1 that is another marker of nephrotoxicity in addition to B2MG, discussed in  the paper. There are also many other markers of nephrotoxicity, e.g. KIM-1, osteopontin, fatty acid binding protein…I believe the discussion would gain value if the authors also referred to these markers. Are there studies in the literature of cadmium induced nephrotoxicity using these proteins in assessing kidney damage?

8. I wonder why the authors did not assess the concentration of cystatin C in studied patients? Cystatin C is also a marker of GFR and it is considered to be even better marker of glomerular filtration compared to creatinine. Thus, there is a question why the urinary cadmium excretion rate has not been also assessed in relation to cystatin C?

Author Response

Response to Reviewer 2

We thank the Reviewer for his/her evaluation of our work and for comments to improve our paper. We have revised our paper accordingly and provide below point-by-point response to each comment. Changes to the text are in blue.

Comments and Suggestions for Authors

Review of the manuscript IJERPH-1868859

Dose-Response Analysis of the Tubular and Glomerular Effects of Chronic Exposure to Environmental Cadmium

by Satarug et al.

The evaluated manuscript is about the assessment of cadmium nephrotoxicity based on a retrospective evaluation of patients with chronic low and high exposure to environmental cadmium (Cd). The authors aimed to ascertain a relationship between Cd exposure with risk of adverse effects on kidney glomerular and tubular function and to evaluate the clinical usefulness of the normalization of excretion rates of Cd, beta-2-microglobulin (B2MG) and N-acetyl-beta-D-glucosaminidase (NAG) to excretion of clearance of creatinine in the laboratory assessment of cadmium-induced nephrotoxicity. The authors concluded that normalization of urinary excretion of Cd, B2MG and NAG to creatinine clearance. Normalization of urinary excretion of Cd, β2MG and NAG to Ccr strengthened the associations of Cd with all three effect indictors; eGFR, β2MG and NAG excretion rates. Moreover, the Cd toxicity threshold is based on excretion of B2MB > 300 ug/g creatinine. According to Authors, the better endpoint for the Cd-induced nephrotoxicity and health risk calculation is reduction in GFR.

The description and conclusions relating to the nephrotoxicity of cadmium is nothing new and it has been proven many times in toxicology. The novelty of the manuscript may be the authors' proposal to use certain parameters, especially the urinary cadmium excretion normalized to creatinine clearance in estimation of Cd-reduction of GFR.

Response:

We agree with the Reviewer that there are numerous studies repeatedly reported the same effects of Cd exposure on tubular dysfunction based on EB2MG and ENAG normalized to Ecr. However, reports on the glomerular effects of Cd are limited, and three are only a few reports that considered simultaneously tubular and glomerular effects like ours. As shown by results of the present study, the glomerular effect of Cd is not demonstratable when ECd is normalized to Ecr.  The Reviewer has correctly identified that normalization of ECd to Ccr is required to show an impact of Cd on GFR, and this represents novel aspect of our work with public health implications.  We have rewritten part of Conclusion to better reflect the novelty of our work.

Reading the manuscript raised some questions and doubts:

Point 1. Abstract of the paper should mention the methodology of work to a greater extent; one may wrongly get the impression that the results were obtained in a clinical trial and not an analytical, retrospective study.

Response: 

We have rewritten part of Abstracts that address the Reviewer’s concerns (lines 12-19).

Point 2. Introduction lines 67-70 - one of the formulas for estimating GFR was mentioned (CKD-EPI formula). At this point, it is also worth mentioning other formulas that enable the estimation of GFR. What about the MDRD formula; why authors decided to use the CKD-EPI formula and not MDRD one?

Response:

We have rewritten part of Introduction. Our justification of CKD-EPI equations is provided in subjection 2.3. Estimated Glomerular Filtration Rate (lines 109-113).

Point 3. Materials and Methods - reading this section may give rise to the erroneous impression that the authors conducted their own research, while their research is a retrospective study and it is based on a population previously enrolled in other clinical study. Thus, the title of the source study, the years of the project's implementation, and description the study protocol should be provided in more detail at the beginning of this section.

Response: 

We have rewritten part of section 2.1. Study Subjects to reflect the nature of our study as advised (lines 70-72).

  1. Materials and Methods – 2.2. Collection of biological specimens and analyses - urinary cadmium concentrations in the two populations distinguished on the basis of expected cadmium exposure (low vs. high) were determined by two different analytical methods. I wonder why the authors did not measure the cadmium concentration in the urine of patients from the area with high cadmium exposure also using the ICP / MS method, that is regarded to be more much more precise and accurate compared to atomic absorption spectrophotometry?

Response:

A very high cost of ICP/MS analysis per se, and an additional requirement for shipment of samples from Thailand to Australia for ICP/MS analysis were prohibitive factors. Also, we based our decision on urinary levels of Cd reported by others for residents of a high-Cd exposure area which could be reliably quantified by the atomic absorption spectrophotometry.

Point 5. In the Results chapter and in Table 1, the authors presented the summary characteristics of the patients. In my opinion, a comparison of the two subpopulations should also be introduced. The research covered both people living in areas with low and high environmental exposure to cadmium. Thus, both groups should be characterized, not just the aggregate population.

Response: 

To achieve sufficient statistical power and certainty, we analyzed the aggregate population. As advised, a new “Table 2. Characteristics of study subjects stratified by residential locality” has been added together with a new subsection, 3.2 3.2. Effects of Residential Location and Smoking on Cadmium Excretion Levels to fully address the issues raised (lines 167-186).

Point 6. Materials and Methods – line 80 - what other drugs (if any) were used by the study subjects, except those ones used to treat hypertension or diabetes?

Response:  

A regular use of any drugs was an exclusion criterion to ascertain that recruited subjects could be considered as apparently healthy persons.  To the best of our knowledge the most commonly used drug was paracetamol.

Point 7. Discussion – lines 318-320 – Authors mentioned neutrophil gelatinase-associated lipocalin NGAL-1 that is another marker of nephrotoxicity in addition to B2MG, discussed in  the paper. There are also many other markers of nephrotoxicity, e.g. KIM-1, osteopontin, fatty acid binding protein…I believe the discussion would gain value if the authors also referred to these markers. Are there studies in the literature of cadmium induced nephrotoxicity using these proteins in assessing kidney damage?

Response: 

Thank you for the suggestion.  We have added information on KIM1 as a sensitive marker of injury to kidney tubular cells caused by Cd (lines 344-351).

Point 8. I wonder why the authors did not assess the concentration of cystatin C in studied patients? Cystatin C is also a marker of GFR and it is considered to be even better marker of glomerular filtration compared to creatinine. Thus, there is a question why the urinary cadmium excretion rate has not been also assessed in relation to cystatin C?

Response:

We will consider an analysis of cystatin C in our future study to strengthen our proposition of using GFR reduction as an endpoint for risk assessment of Cd exposure.  In the present work, we used creatinine-based eGFR equations of the CKD-EPI because this has been in clinical to follow the progression of kidney disease from stage 1 to stage 5 (lines 117-119).

Round 2

Reviewer 2 Report

Re-review of the manuscript IJERPH-1868859

Dose-Response Analysis of the Tubular and Glomerular Effects of Chronic Exposure to Environmental Cadmium

by Satarug et al.

I would like to thank the Authors for responding to my comments in the form of a point-to-point response and for sending a new version of the manuscript.

I think that after the introduced amendments, the work gained in an overall quality.

1. The another thing that I would like to suggest to the Authors is a broader reference to their own, recently published paper:

Satarug S, Vesey DA, Nishijo M, Ruangyuttikarn W, Gobe GC, Phelps KR. The Effect of Cadmium on GFR Is Clarified by Normalization of Excretion Rates to Creatinine Clearance. Int J Mol Sci. 2021 Feb 10;22(4):1762. doi: 10.3390/ijms22041762. PMID: 33578883; PMCID: PMC7916559.

This paper is cited in the present manuscript as [16]. Even though it is thematically and methodologically similar to the subject of the present manuscript and one may even get the impression of an unauthorized repetition, the Authors do not pay much attention to their own previous research (the only reference is verses 59-62). I think it would be useful to discuss the results discussed in the present manuscript to a greater extent and to relate them to those ones given in the article already published in Int J Mol Sci. In this way, the present discussion could be an overview of these both articles in the form of an “interrelated publications”.

2. Final remark: 9 items cited in References are self-citations (out of 39 of the total items) – [1], [2], [3], [9], [11], [15], [16], [17], [35].

Perhaps, it would be "safer" to replace at least the first three items from the Introduction with other citations to avoid the accusation of excessive self-citations? Of course, this is just a “non-binding” suggestion.

Author Response

Response to additional comments

We thank the Reviewer for his/her thorough review of our work and for another opportunity to further improve our manuscript.  We have revised our paper accordingly and provide below responses to the issue raised. Changes to the text are in blue.

Comments and Suggestions for Authors

Re-review of the manuscript IJERPH-1868859

Dose-Response Analysis of the Tubular and Glomerular Effects of Chronic Exposure to Environmental Cadmium

by Satarug et al.

I would like to thank the Authors for responding to my comments in the form of a point-to-point response and for sending a new version of the manuscript.

I think that after the introduced amendments, the work gained in an overall quality.

  1. The another thing that I would like to suggest to the Authors is a broader reference to their own, recently published paper:

Satarug S, Vesey DA, Nishijo M, Ruangyuttikarn W, Gobe GC, Phelps KR. The Effect of Cadmium on GFR Is Clarified by Normalization of Excretion Rates to Creatinine Clearance. Int J Mol Sci. 2021 Feb 10;22(4):1762. doi: 10.3390/ijms22041762. PMID: 33578883; PMCID: PMC7916559.

This paper is cited in the present manuscript as [16]. Even though it is thematically and methodologically similar to the subject of the present manuscript and one may even get the impression of an unauthorized repetition, the Authors do not pay much attention to their own previous research (the only reference is verses 59-62). I think it would be useful to discuss the results discussed in the present manuscript to a greater extent and to relate them to those ones given in the article already published in Int J Mol Sci. In this way, the present discussion could be an overview of these both articles in the form of an “interrelated publications”.

Response:

  • Thank you for raising this important issue. We have elaborated that the present work contributes to new knowledge while confirming results reported in our previous publication; Int. J. Mol Sci. 2021 Feb 10;22(4):1762 (lines 351-365). Two new relevant references are added (ref. 30, ref. 36).

  1. Final remark: 9 items cited in References are self-citations (out of 39 of the total items) – [1], [2], [3], [9], [11], [15], [16], [17], [35].

Perhaps, it would be "safer" to replace at least the first three items from the Introduction with other citations to avoid the accusation of excessive self-citations? Of course, this is just a “non-binding” suggestion.

Response:

  • To address, the Reviewer’s concern, we retained our own ref. 1 which now become ref 4. Our own former ref. 2 and ref. 3 are replaced by three recent cadmium reviews by three different research groups others as suggested (Lines 34-36).

  • In addition, we have made further improvement by adding fundamental knowledge on the utility of an increased β2M as tubulopathy marker (lines 45-50) with 3 new references (ref. 7-9).

  • In total, 45 publications are cited.

Ref.1 Nordberg, M.; Nordberg, G.F. Metallothionein and cadmium toxicology-historical review and commentary. Biomolecules 2022, 12, 360.

Ref 2. Genchi, G.; Sinicropi, M.S.; Lauria G, Carocci A, Catalano A. The effects of cadmium toxicity. Int. J. Environ. Res. Public Health 2020,17, 3782.

Ref. 3 Thévenod, F.; Lee, W.K.; Garrick, M.D. Iron and cadmium entry into renal mitochondria: Physiological and toxicological implications. Front. Cell Dev. Biol. 2020, 8, 848.